# SlimLLaVA: Automatic Pruning for Large Vision-language Models

## Abstract

Multimodal large language models achieve satisfying performance in complex reasoning tasks, while still suffers from high model complexity in deployment especially for resource-limited devices. In this paper, we propose an automatic pruning method of large vision-language models for efficient multimodal reasoning. Conventional methods leverage the training data of the original model to select the proper pruning ratio for different network components, while they are infeasible for large vision-language models due to the unbearable search cost caused by web-scale training corpus. On the contrary, we only use a few samples to search the desired pruning policy by maximizing its generalization ability on the unknown training data despite of the model accuracy, so that the optimal accuracy-efficiency trade-off can be obtained for large vision-language models. Specifically, we formulate the generalization gap for the pruning policy based on the structural risk minimization principle. With the task performance and the generalization ability, we iteratively search for the optimal pruning policy in the given search space and optimize the vision projector to evolve the search space with higher upper bound of performance. We conduct extensive experiments on ScienceQA, Vizwiz, MM-vet and LLaVA-Bench datasets for the task of visual question answering. With only 64 samples for pruning policy search, our method achieves 83.05% accuracy on ScienceQA and $\times 1.47$ speedup compared to the dense LLaVA-v1.5-7B model.

## 1 Introduction

Multimodal large language models have been widely adopted in complex reasoning tasks such as visual question answering (Shao et al., 2023; Guo et al., 2023), embodied task planning (Huang et al., 2022; Mendez-Mendez et al., 2023) and dialogue systems (Li et al., 2018). Although they achieve excellent performance in the high-level tasks with rich commonsense, the inference stage of large vision-language models (LVLMs) in deployment requires large memory footprint and long latency because of the tremendous neurons. In order to deploy the powerful LVLMs on mobile devices such as cellphones, navigation robots and autonomous vehicles with strict computational resource limit, we are required to reduce the model complexity without obvious degradation on the performance.

Network pruning aims to remove unimportant components including layers, channels, tokens and neurons to reduce the storage and computational complexity without harming the task performance, which have been widely studied for computer vision (Burtsev et al., 2018; Kong et al., 2022), natural language processing (Sanh et al., 2020; Ma et al., 2023) and system control (Tang et al., 2020). The components importance can be defined as the $l_1$ and $l_2$ magnitude of weights and activations (Sun et al., 2023), and the Jacobian and Hessian matrix regarding the loss functions (Qiao & Yoo, 1999). Since different components usually have various sensitivity to pruning in the downstream tasks, automatic pruning assigns the optimal pruning ratio to each component in order to achieve higher accuracy-efficiency trade-off. In conventional methods, the training data of the original model is leveraged to evaluate the sampled pruning policy based on the accuracy and the model complexity, which provides feedback to the search algorithm to select better candidates. However, LVLMs require extensive training corpus, and the search cost associated with existing automatic pruning methods is prohibitively high.

In this paper, we present SlimLLaVA that automatically prunes LVLMs to achieve low model complexity and unaffected task performance with acceptable search cost. Unlike existing automatic pruning methods that leverage the large-scale training data of the original models for pruning policy search, our method only use a few samples to assign the optimal pruning ratio for weight matrix in each layer. Since we maximize the generalization ability of the pruning policy to the unknown web-scale training corpus, the acquired solution achieves satisfying trade-off between accuracy and efficiency in a wide variety of downstream tasks. More specifically, we first formulate the gener-

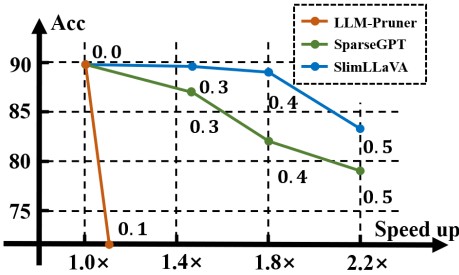

Figure 1: The trade-off between accuracy and inference speed on ScienceQA with Slim-LLaVA.

alization gap of the pruning policy between the proxy samples for policy search and the unknown training set via structural risk minimization principle, which can be evaluated by the Frobenius norm of the weight matrix in the pre-trained LLMs. We then evolve the pruning policy candidates with the goal of achieving higher task performance and generalization ability, where the weights of MLP and attention are pruned based with the selected pruning ratio. The vision projector is optimized to enhance the upper bound of the accuracy and the generalization ability for all policies in the pruning space, which is estimated via the Euclidean distance between the sampled policy and neighbor candidates. The policy candidate selection and the pruning space evolution are implemented iteratively to acquire satisfying accuracy-efficiency trade-off in the extremely large pruning space with acceptable cost. Figure 1 demonstrates the trade-off between accuracy and inference latency on ScienceQA for uniform pruning methods SparseGPT (Frantar & Alistarh, 2023) and our SlimLLaVA. We conducted extensive experiments in a wide variety of multimodal reasoning tasks including short-answer, option-only for multiple-choice and natural QA, and the results shows that SlimLLaVA can achieve 83.05% accuracy on ScienceQA and ×1.47 speedup compared to the dense LLaVA-v1.5-7B model. Our contributions can be summarized as follows:

- We introduce SlimLLaVA, a novel pruning method for large vision-language models that effectively reduces model complexity while preserving task performance with minimal data usage.

- Our approach leverages the structural risk minimization principle to maximize the generalization ability of pruning policies, enabling efficient adaptation to unknown training data.

- Extensive experiments demonstrate that SlimLLaVA achieves a significant speed-up in inference while maintaining competitive accuracy across various multimodal reasoning tasks.

## 2 RELATED WORK

**Large Vision-language Models:** Large Vision-language Models(LVLMs) fuse visual and linguistic modals via bridging the gap between different modalities that achieved outstanding performance such as in-context predictions (Liu et al., 2023; Salewski et al., 2024), multi-image and chain-of-thought reasoning (Yang et al., 2023; Driess et al., 2023). To utilize learnable connector module projecting visual information into language modal space, token-level fusion and feature-level fusion are presented for encoding multimodal information to LLMs. For token-level fusion, LLaVA (Liu et al., 2024b) series utilized linear MLPs to align the image tokens with language tokens, finetuning LVLMs with visual instruction samples to enhance zero-shot capabilities. MM1 (McKinzie et al., 2024) compared the significance of varied architectural aspects and found the number of visual tokens and input resolution was more important than the design of projector. For feature-level fusion, Flamingo (Alayrac et al., 2022) inserted extra cross-attention layers to connect pre-trained vision-only and language-only model, significantly boosting learning efficiency. CogVLM (Wang et al., 2024) integrated a visual expert module into every Transformer layer to fuse the vision and language features. Existing methods usually focus on the performance of LVLMs, ignoring the real-world deployment particularly in environments with limited resources. Despite light-weight LVLMs such as MobileVLM (Chu et al., 2023), TinyLLaVA (Zhou et al., 2024) and quantinized methods such as GPTQ (Frantar et al., 2022), AWQ (Lin et al., 2023) have reduced the size of LMMs, the cost still exceeds the resource constraint of mobile devices and they fail to consider the intermodal interaction. Differently, our method focus on pruning the pre-trained LVLMs to find the optimal trade-off between accuracy and efficiency with few-shot proxy samples.

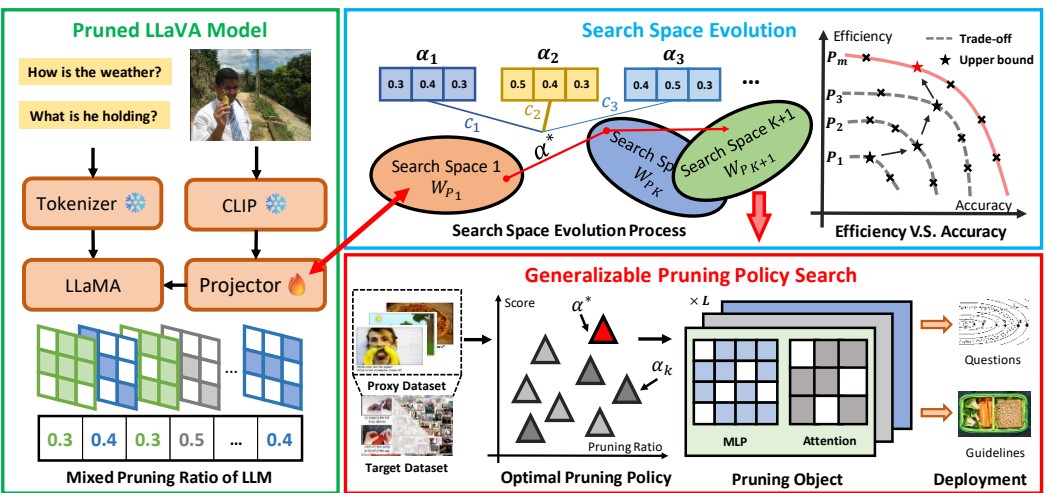

Figure 2: The overall pipeline of SlimLLaVA. In each iteration, we first search for the optimal pruning policy for matrix in each LLaMA layer, where evolutionary algorithms are employed with the fitness function containing model accuracy and generalization ability. Then we evolve the search space by optimizing the projector weight so that the upper bound of accuracy and generalization ability for all policies can be improved.

**Automatic Pruning:** Automatic pruning(Luo & Wu, 2020; Lin et al., 2020) removes unnecessary connections, neurons, and layers to reduce the model size, while the model accuracy remains high. The early work of automotic pruning AMC (He et al., 2018) utilized reinforcement learning to effectively explore the design space automatically, thereby enhancing the quality of model compression. To achieve fine-grained pruning across samples, dynamic pruning is proposed to predict an instance-wise pruning strategy for each input sample. AutoPrune (Xiao et al., 2019) pruned the network through optimizing a set of trainable parameters instead of original weights. DynamicVit (Rao et al., 2021) progressively and adaptively removed unnecessary tokens based on the input due to the sparse attention. AutoCompress(Liu et al., 2020) introduces an automatic structured pruning framework that improves model compression and inference speed using ADMM-based pruning and heuristic search, achieving high pruning rates. When pruning large pre-trained neural networks, the training set for the original model cannot be acquired. To address this, post-training pruning(Kwon et al., 2022; Lazarevich et al., 2021) derives the sensitivity of each component to pruning by first-order and second-based gradient information. Dynamic Context Pruning (Anagnostidis et al., 2024) utilized a learnable mechanism that selects uninformative tokens to be removed from the context at various stages of the generation process. However, traditional automatic pruning methods are not satisfied for LVLMs due to ignoring the generalization gap between training data and few-shot proxy samples. Different from the previous works, our method evolve the search space to find the optimal pruning policy maximizing the generalization ability.

## 3 APPROACH

In this section, as illustrated in Figure 2, we first introduce the preliminary of automatic pruning for LVLMs. We then formulate the generalization gap between the few-shot proxy samples for pruning policy search and the overall data distribution, and search the pruning policy with generalizability maximization in given search space. Finally, we evolve the search space to improve the upper bound of accuracy and generalizability ability for all policies. We ensure that the search space remains flexible and evolves dynamically with each iteration, improving the overall pruning performance.

### 3.1 AUTOMATIC PRUNING FOR LVLMS

Model pruning aims to remove unnecessary or redundant elements to reduce the size of neural network and computational complexity, while minimizing the impact on performance. Automatic pruning assigns optimal pruning ratio to different components to achieve the optimal trade-off

between accuracy and efficiency. However, conventional automatic pruning methods (Gao et al., 2022; Anagnostidis et al., 2024) search the pruning policy on the large-scale training data, which is impractical for LVLMs due to the prohibitive training cost and the lack of the training corpus. The goal of automatic model pruning is to minimize loss objective with the sparsity constraint:

$$\min_{\boldsymbol{\alpha}} \ L_{train}\big(\hat{\mathbf{M}}(\boldsymbol{\alpha}), \boldsymbol{\alpha}\big)$$

$$s.t. \quad ||\boldsymbol{\alpha}||_1 = C_0, \ \ \hat{\mathbf{M}}(\boldsymbol{\alpha}) = \arg\min \ L_{train}(\mathbf{M}, \boldsymbol{\alpha}) \tag{1}$$

where $L_{train}$ represents the loss in the large-scale training set for LVLMs. $\boldsymbol{\alpha}$ refers to the pruning policy, which determines how much of each layer or component to prune. The pruning policy $\boldsymbol{\alpha}$ denotes sparsity ratio for weight matrix in each layer, and $||\cdot||_1$ means the $L_1$ norm. This ensures that a certain proportion of the model's parameters are removed (pruned) while minimizing the impact on accuracy. $C_0$ stands for the overall sparsity limit for the entire LVLM. The optimal pruning masks for weights $\hat{\mathbf{M}}(\boldsymbol{\alpha})$ are acquired by minimizing the loss on the training data. Due to the limited proxy samples during pruning policy search of pre-trained LVLMs, the acquired policy usually overfits the proxy data without generealization to the entire dataset. Therefore, we should also consider the generalization gap between the provided proxy samples and the real training data distribution for the pruning policy, and we can obtain pruning policies with satisfying accuracy-efficiency trade-off in diverse downstream tasks.

## 3.2 GENERALIZABLE PRUNING POLICY SEARCH FOR LVLMS

Evaluating the true network objective for searched pruning policies is very challenging because of the limited samples compared with the entire training data. Therefore, we maximize the generalization ability of the pruning policy via the structural risk minimization (SRM) principle, which can be bounded by the empirical risk and the unseen data distribution. Structural Risk Minimization (SRM) is a principle from statistical learning theory that seeks to balance model complexity with training data performance, minimizing the risk on both known and unseen data. The optimal trade-off between complexity and model performance ensures that our approach can be applied effectively in various real-world applications. For a dataset containing N samples in pruning policy search, the structural risk is written in the following with the probability at least $1 - \epsilon$:

$$\mathbb{E}_r[L(\mathbf{M}, \boldsymbol{\alpha})] \leqslant \mathbb{E}_e[L(\mathbf{M}, \boldsymbol{\alpha})] + 2\mathcal{R}(\mathcal{F}) + \sqrt{\frac{\ln 4/\epsilon}{N}} \tag{2}$$

where $\mathbb{E}_r$ and $\mathbb{E}_e$ are the expectation over the unknown real data distribution and the empirical expectation over limited proxy samples for policy search. $\mathcal{R}(\mathcal{F})$ means the Rademacher complexity over the objective function class $\mathcal{F}$. A lower Rademacher complexity indicates better generalization on unseen data. Since the empirical risk represents the performance of the pruning policy on the limited provided samples, we minimize the Rademacher complexity to reduce the gap between the empirical risk and the real structural risk. The Rademacher complexity means the upper bound of the weighted sum of objectives across samples for all possible objectives. Following (Neyshabur et al., 2015), we efficiently evaluate the Rademacher complexity via the matrix norm for all layers in the model:

$$\mathcal{R}(\mathcal{F}) = \frac{1}{N} \cdot \mathbb{E}_\sigma \left[ \sup_{L \in \mathcal{F}} \sum_{i=1}^{N} \sigma_i L_{x_i}(\mathbf{M}, \boldsymbol{\alpha}) \right] \propto \eta \prod_{i=1}^{K} \mathcal{A}_i \tag{3}$$

where $L_{x_i}$ represents the loss function of the $i_{th}$ sample $x_i$ for policy search, and $\{\sigma_i\}$ are independent random variables drawn from the Rademacher distribution. $\mathcal{A}_i$ stands for the Frobenious norm of the parameter matrix in the $i_{th}$ layer, and $\eta$ can be regarded as a constant related to the layer number and the sample size. Low weight norm indicates that the model output is weakly correlated with the input, which leads to high generalization ability for different data distribution. Despite of selecting components with satisfying accuracy-efficiency trade-offs on the limited proxy samples, we also encourage the remaining components to have low weight norm. Therefore, the acquired pruning policy can be adapted to downstream tasks with unseen diverse data distribution.

Since the LLaMA model in multimodal LLMs contribute significantly to the overall model complexity, we prune the weight matrix for each MLP in the LLaMA model. The reason for excluding the visual encoder and projector from pruning is that pruning these components minimally impacts computation

but can cause significant performance degradation. The $k_{th}$ element of $\boldsymbol{\alpha}$ represents the pruning ratio of the $k_{th}$ MLP layer. The fitness function $J$ for generalizable search can be written as follows:

$$\max_{\boldsymbol{\alpha}} \quad J(\boldsymbol{\alpha}) = \mathbb{E}_e[L(\mathbf{M}, \boldsymbol{\alpha})] + \eta \prod_{i=1}^{K} \mathcal{A}_i(\mathbf{M}_i, \alpha_i) \tag{4}$$

where $\mathcal{A}_i(\mathbf{M}_i, \alpha_i)$ means the matrix norm of the $i_{th}$ layer for the pruning ratio $\alpha_i$ and the pruning mask $\mathbf{M}_i$. Given the pruning ratio for each layer, we acquire the pruning mask by OBS algorithm (Frantar & Alistarh, 2023) removing weights that have lowest influence on the layer output. The OBS (Optimal Brain Surgeon) algorithm optimally removes weights while minimizing the increase in loss, thus preserving model performance. We employ the evolutionary algorithms to search for the optimal pruning policy.

### 3.3 SEARCH SPACE EVOLUTION OF PRUNING POLICIES FOR LVLMS

LVLMs align the visual tokens with the language tokens via the projector layer, so that the knowledge learned in the pre-trained CLIP visual encoder and the LLaMA model can be fully leveraged for multimodal reasoning. Therefore, we can optimize the weights of the projector, where the possible highest accuracy and generalizability can be acquired for the pruning policies given the resource budget. In this context, the projector plays a crucial role in fusing visual and language modalities, making it a key target for optimization in ensuring high model performance even after pruning. The projector weights are regarded as the search space of the pruning policy search in LVLMs, and optimizing the projector weights is equivalent to search space evolution. The objective for the evolution is as follows:

$$\max_{\boldsymbol{W}_p} \sup_{\boldsymbol{\alpha}} J(\boldsymbol{W}_p, \boldsymbol{\alpha}) = \sup_{\boldsymbol{\alpha}} \left[ \mathbb{E}_e[L(\boldsymbol{W}_p, \mathbf{M}, \boldsymbol{\alpha})] + \eta \prod_{i=1}^{L} \mathcal{A}_i(\mathbf{M}, \boldsymbol{\alpha}) \right] \tag{5}$$

where $J(\boldsymbol{W}_p, \boldsymbol{\alpha})$ means the fitness function with respect to the projector weight $\boldsymbol{W}_p$ and the pruning policy $\boldsymbol{\alpha}$, and $L(\boldsymbol{W}_p, \mathbf{M}_w, \alpha)$ means the loss function of the model regarding the projector weight despite of the pruning mask and the pruning policy. The optimization goal here is to find the best combination of projector weights and pruning ratios to maximize performance. This step ensures that the search space continually adapts and improves, rather than being static. By maximizing the upper bound of the fitness function with search space evolution, pruning policies with higher performance and generalizability appear in the evolved search space. However, estimating the upper bound of the fitness function over all pruning policies is infeasible as numerating the countless selections is computationally prohibited. However, calculating the upper bound for all pruning policies is computationally infeasible. We relax the original problem as the following one with the weighted fitness function $\hat{J}$:

$$\max_{\boldsymbol{W}_p} = \hat{J}(\boldsymbol{W}_p, \boldsymbol{\alpha}) = \mathbb{E}_e[c_m L_m(\boldsymbol{W}_p, \mathbf{M}, \boldsymbol{\alpha})] \tag{6}$$

where $c_m$ and $L_m$ represent the importance weight and the loss function for the $m_{th}$ candidate pruning policy in the current search space. The candidate that is closer to the upper bound should be considered with higher importance in the search space evolution, and the corresponding importance weight should be larger. Meanwhile, the matrix norm is independent of the projector weights, and it is omitted in the optimization objective of the projector weights. The importance weight is estimated via the following rule:

$$c_m = \frac{\exp(-\sum_{\boldsymbol{\alpha} \in \mathcal{N}_m} ||\boldsymbol{\alpha}_m - \boldsymbol{\alpha}||^2)}{\sum_k \exp(-\sum_{\boldsymbol{\alpha} \in \mathcal{N}_k} ||\boldsymbol{\alpha}_k - \boldsymbol{\alpha}||^2)} \tag{7}$$

where $\mathcal{N}_m$ means the neighborhood of the $m_{th}$ candidate pruning policy defined by Euclidean distance. The normalized importance weight $c_m$ assigns higher values to candidates with closer neighbors, implying that candidates in dense regions of the search space (close to other candidates) are considered more important. Different candidates tend to converge towards the optimal pruning policy, and a policy with neighbors at shorter distances suggests a higher likelihood of being optimal, as it is closer to the optimal pruning solution. This weight calculation ensures that pruning policies with stronger potential for optimality are prioritized during the search process, improving the overall quality of the evolved search space.

---

**Algorithm 1:** Search Space Evolution of Generalizable Pruning Policy

---

**Input** : Initial search space $\boldsymbol{W}_p^0$, Pruning ratio constraint $C$, max evolution step $\tau$, the number
     of pruning policy sampling $n$, the size of evolution group
**Output** : The optimal pruning ratio $\boldsymbol{\alpha}$, search space $\boldsymbol{W}_p$

1  $G_0 :=$ Randomly sample $n$ pruning policy $\{\boldsymbol{\alpha}_1, \boldsymbol{\alpha}_2, ..., \boldsymbol{\alpha}_n\}$ with the constraints $\mathcal{C}$
2  **while** *evolution step $t \leqslant \tau$* **do**
3  |    Prune networks base on the OBS pruning method and pruning policy $G_t$ in the $t_{th}$ round
4  |    Obtain top-$k$ candidates with fitness function (4)
5  |    Generate $G_{t+1}$ accoding to evolutionary algorithm
6  |    Predict the optimal pruning policy via Euclidean distance in (7)
7  |    Evolve the search space $\boldsymbol{W}_p^t$ base on (6)
8  **end**

---

The overall pipeline for automatic pruning of LVLMs is demonstrated in Algorithm 1. For a pre-trained multimodal LLM, we first search for the optimal pruning policy with the limited proxy samples given the search space represented by the projector weights. Then we remove the unimportant weights for each layer with the searched pruning ratio via the OBS pruning method. Finally, we optimize the projector weight for search space evolution. The above three stages are iteratively implemented in each round until reaching the search cost constraint.

## 4  EXPERIMENTS

In this section, we conducted extensive experiment for LLaVA architectures on multi-modal question answering datasets. Firstly, we introduce the implementation details of our SlimLLaVA and the dataset information. Then we conduct ablation study to evaluate the effectiveness of the generalizable pruning policy search and the search space evolution of pruning policies. Finally, we compare the performance regarding accuracy and efficiency with existing pruning methods.

### 4.1  SETUPS

**Dataset:** We conduct the experiments on ScienceQA (Lu et al., 2022), Vizwiz (Bigham et al., 2010), MMVet (Yu et al., 2023) and LLaVA-Bench (Liu et al., 2024b). ScienceQA is the first large-scale multimodal dataset that annotates lectures and explanations for the answers containing 21k multiple-choice science questions. Moreover, ScienceQA dataset encompasses vision-language pairs that can be categorized into subjects such as natural science (NAT), social science (SOC), and language science (LAN). Additionally, another classification based on context modality includes text context (TXT), image context (IMG), and no context (NO). ScienceQA dataset stands out for having the largest collection of images, covering all 12 grades, containing the longest questions, and featuring the most diverse input sources. Vizwiz arose from a natural setting for visual question answering with blind individuals, together with 10 crowdsourced answers per visual question, contributing images and spoken questions, which contains 20k image-question pairs. MMVet leverages the amalgamation of various core visual-linguistic competencies to address complex challenges which defines 16 emergent tasks of interest integrated from the six defined core VL capabilities. LLaVA-Bench is a benchmark designed to test how well multimodal AI models can handle real-world visual tasks by engaging in open-ended conversations based on diverse images. These datasets present distinct challenges, from the complexity of real-world queries to specialized reasoning tasks, ensuring that SlimLLaVA's performance is tested under a variety of realistic conditions.

**Implementation Details** We evaluate our automatic pruning method for pre-trained LVLMs including LLaVA-SQA-7B (Liu et al., 2024b), which is finetuned on ScienceQA dataset and LLaVA-v1.5-7B (Liu et al., 2024a), which achieves SoTA on a broad range of 11 tasks. The average pruning ratio $p_0$ for the entire model is set to 0.3, 0.4 and 0.5. To ensure the average pruning ratio of all candidate pruned networks remains consistent, the choices of pruning ratio for each layer is selected from $[p_0 - 0.1 : 0.05 : p_0 + 0.1]$. This ensures the pruning process adapts to the structure of each layer without significantly affecting the model's integrity. We also set the upper and lower bounds for the generated candidate pruning ratios as $p_0 + 0.01$ and $p_0 - 0.01$. This range is carefully chosen

Table 1: Comparison of average pruning ratio and the accuracy evaluated on ScienceQA across different pruning method.

| Criteria | Avg ratio | Accuracy |
|---|---|---|
| Uniform | 0.50 | 80.45 |
| Auto w/o Gen | 0.49 | 81.34 |
| SlimLLaVA | 0.50 | 83.05 |

Table 2: The effects of search space evolution regarding the average pruning ratio and the accuracy.

| technique | Avg ratio | Accuracy |
|---|---|---|
| No Evo | 0.50 | 81.07 |
| Evo w/o UB | 0.51 | 82.85 |
| SlimLLaVA | 0.50 | 83.05 |

to allow sufficient flexibility for pruning each layer while maintaining overall consistency with the target pruning ratio. For proxy data, we use 64 segments with 256 tokens for each segment which are randomly sampled from the PTB dataset (Marcus et al., 1993) which is versatile and has been used in various tasks, covers a wide range of English language constructs, making it a valuable resource for understanding and modeling the syntactic and semantic aspects of the language. It embodies generic data extracted from internet sources, which indicates the zero-shot settings of our experiments because no task-specific data is provided. We evolve 16 policy candidates in each round, and the value of the hyperparameter $\eta$ in LLaVA-SQA-7B pruning is set to 0.5. This number of policy candidates ensures sufficient exploration of the pruning space without incurring excessive computational overhead. Given the searched pruning ratio, we use the same pruning methods as in SparseGPT (Frantar & Alistarh, 2023) for model pruning to fully leverage the potential in network pruning with high degrees of freedom. We also sparsify Transformer layers sequentially and only prunes the matrix of attention weights and MLP weights in the pre-trained LLaMA model. During search space evolution, we select top-5 nearest neighbors to compute the importance weight in (7). The number of rounds containing policy search, weight pruning and search space evolution is 10.

## 4.2 ABLATION STUDY

**Effects of generalizable automatic pruning:** Automatic pruning determines optimal pruning ratios for various components to achieve the ideal trade-off between accuracy and efficiency. In order to minimize the generalization gap between the provided samples and training data distribution, we significantly enhance the generalization ability of the pruning policy via the structural risk minimization (SRM) principle. The use of SRM allows us to balance the pruning policy across layers, taking into account both the accuracy on the proxy samples and the unseen data. This leads to the optimal pruning policy, especially on challenging tasks where overfitting to the proxy samples could otherwise degrade performance. We compare our pruning method with uniform pruning policies (Uniform) and that without considering the generalization ability (Auto w/o Gen). Table 1 shows the average pruning ratio of the pruned model and the corresponding accuracy on ScienceQA, where both the automatic pruning and generalization improvement significantly contribute to the accuracy-efficiency trade-off on the acquired pruning policy. Uniform pruning policy ignores the layer-wise importance variance of pre-trained LVLMs, and searching the pruning policy on limited proxy samples without considering the generalization gap substantially degrades the performance of the policy on downstream tasks due to the overfitting.

**Influence of number of proxy samples and token length:** The pruning results and training cost are influenced by the number of proxy samples $n$ and the length of tokens $s$ during the pruning process. To find the optimal trade-off between the accuracy-efficiency trade-off and the search cost, we varied the value of $n$ and $s$ to observe their effects on the pruning outcomes and computational cost. Figure 3 demonstrates the results where the pruning ratio constraint was set to 0.4. Increasing the number of proxy samples and the token lengths can both improve the accuracy given the resource budget, while also brings significant search cost burdens to the automatic pruning framework. Meanwhile, the marginal improvement becomes negligible when the number

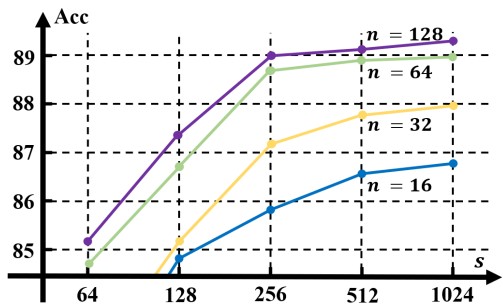

Figure 3: The performance variation with the number of sampled data $n$ and sequence length $s$ in pruning space evolution tested on ScienceQA.

of proxy numbers and the token length exceed

Table 3: The accuracy on ScienceQA for LLaVA-SQA-7B and LLaVA-v1.5-7B, where we evaluate the models across a wide range of domains. Since LLM-Pruner fully loses its question-answering inference capacity when the pruning ratio is above 0.4, comparisons for a pruning ratio of 0.5 are not considered.

| | Ratio | Method | Subject | | | Context Modality | | | Grade | | Average |
|---|---|---|---|---|---|---|---|---|---|---|---|
| | | | NAT | SOC | LAN | TXT | IMG | NO | G1-6 | G7-12 | |
| LLaVA-SQA-7B | 0.0 | - | 89.39 | 96.06 | 85.64 | 88.71 | 87.65 | 88.50 | 90.93 | 87.80 | 89.91 |
| | 0.3 | LLM-Pruner | 39.34 | 27.22 | 47.55 | 39.69 | 34.56 | 43.76 | 38.22 | 40.21 | 38.93 |
| | | SparseGPT | 88.28 | 95.16 | 85.45 | 87.39 | 86.71 | 88.01 | 89.54 | 88.00 | 88.99 |
| | | Ours | 89.39 | 93.36 | 86.45 | 88.51 | 86.47 | 89.34 | 89.98 | 88.53 | 89.46 |
| | 0.4 | LLM-Pruner | 0.18 | 0.11 | 0.64 | 0.20 | 0.15 | 0.49 | 0.18 | 0.46 | 0.28 |
| | | SparseGPT | 80.73 | 89.65 | 79.82 | 80.40 | 80.17 | 81.32 | 83.99 | 79.43 | 82.36 |
| | | Ours | 89.17 | 91.34 | 85.37 | 88.27 | 85.57 | 88.57 | 89.39 | 87.54 | 88.73 |
| | 0.5 | SparseGPT | 80.28 | 89.54 | 73.45 | 79.23 | 78.98 | 77.84 | 82.56 | 76.66 | 80.45 |
| | | Ours | 83.61 | 90.44 | 75.91 | 82.80 | 83.54 | 79.51 | 84.73 | 80.03 | 83.05 |
| LLaVA-v1.5-7B | 0.0 | - | 62.57 | 69.07 | 65.55 | 62.81 | 63.26 | 63.90 | 68.65 | 57.61 | 64.70 |
| | 0.3 | LLM-Pruner | 21.67 | 11.36 | 12.73 | 23.17 | 21.52 | 13.45 | 16.30 | 18.79 | 17.19 |
| | | SparseGPT | 58.88 | 68.05 | 63.45 | 59.24 | 61.87 | 61.39 | 65.12 | 56.36 | 61.99 |
| | | Ours | 64.03 | 67.15 | 69.36 | 63.20 | 62.96 | 68.9 | 68.98 | 60.84 | 66.07 |
| | 0.4 | LLM-Pruner | 0.18 | 0.11 | 0.45 | 0.20 | 0.15 | 0.35 | 0.18 | 0.33 | 0.24 |
| | | SparseGPT | 56.97 | 65.80 | 60.55 | 57.04 | 59.44 | 58.40 | 63.44 | 63.13 | 59.75 |
| | | Ours | 60.35 | 67.27 | 66.91 | 60.26 | 61.77 | 64.53 | 66.74 | 57.68 | 63.50 |
| | 0.5 | SparseGPT | 53.51 | 57.71 | 56.82 | 54.01 | 54.64 | 54.98 | 56.86 | 52.34 | 55.25 |
| | | Ours | 56.57 | 56.13 | 60.55 | 56.79 | 54.24 | 58.54 | 59.73 | 53.53 | 57.51 |

64 and 256 respectively. In order to acquire the satisfying pruning policy with acceptable search cost, we select the number of proxy numbers and the token length to be 64 and 256 for the rest of the experiments. Moreover, we can also optimize the balance between resource consumption and performance, allowing them to meet their specific requirements.

**Effects of search space evolution:** Optimizing the projector weight in LVLMs can further significantly enhance the search space for pruning ratio, where the possible highest accuracy and generalization ability can be achieved. To verify the effectiveness of search space evolution, we conduct the ablation study to evaluate different variants: (a) the method without space evolution (No Evo), (b) the method with space evolution while failing to consider the upper bound in evolution (Evo w/o UB), which means all $c_m$ are set to 1, (c) our method. Table 2 illustrates the performance including the average pruning ratio and the model accuracy. Evoving the search space can enhance the accuracy-efficiency trade-off of the acquired pruning policy, because possible highest accuracy and generalization ability are enhanced. Evaluating the upper bound of the fitness function with the average value degrades the accuracy-efficiency trade-off because of the discrepancy of the optimization objective.

### 4.3 COMPARISON WITH STATE-OF-THE-ART METHODS

In this section, we compare the model pruning techniques in SlimLLaVA with state-of-the-art pruning strategy. As far as we know, we are the first multimodal LVLMs pruning method, and we utilize the strategy designed for LLM pruning including LLM-Pruner (Ma et al., 2023) and SparseGPT (Frantar & Alistarh, 2023) as the baseline method for further comparison. Finally, we provide two visual answers generated by LLaVA-SQA-7B.

**Performance Analysis:** The task of real-world scientific question answering can be leveraged as an effective way to evaluate the performance of large multimodal models. In order to demonstrate our pruning policy applicable to different models, Table 3 illustrates the accuracy on ScienceQA across LLaVA-SQA-7B and LLaVA-v1.5-7B architectures with variable average pruning ratios. LLaVA-SQA-7B first generates a reasoning process based on the given question, and subsequently derives

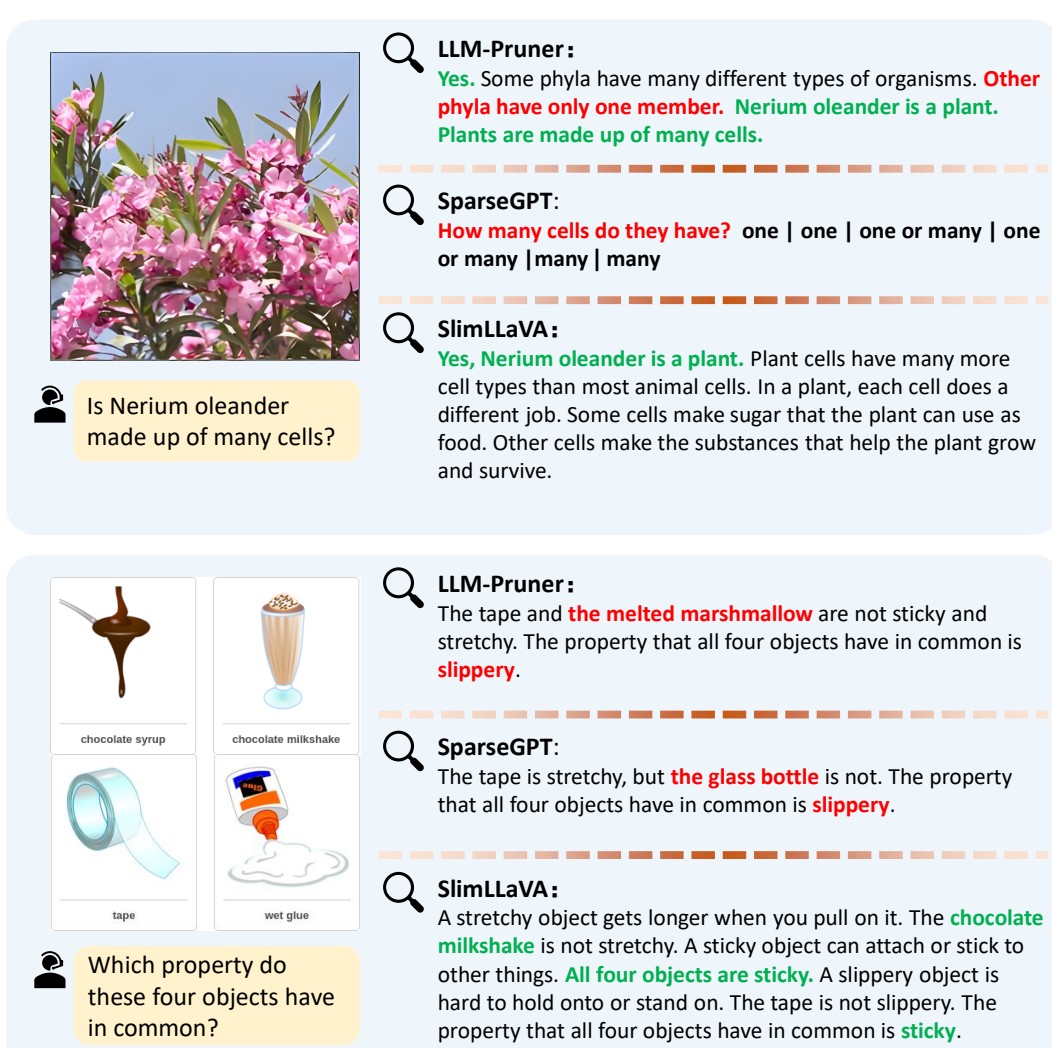

Figure 4: Visual examples from LLaVA-SQA-7B. We color the text to show the response of different pruning methods and SlimLLaVA consistently delivers more refined, contextually appropriate responses, showcasing its superior pruning and reasoning capabilities.

the final answer from this reasoning process. On the contrary, LLaVA-v1.5-7B directly generates the final answer without an intermediate reasoning process. Given that LLM-Pruner completely loses its question-answering inference ability at a pruning ratio above 0.4, we do not make comparisons at a pruning ratio of 0.5. SparseGPT (Frantar & Alistarh, 2023) can reach 50% unstructured sparsity while the accuracy degradation is 9.5%. However, it ignores the significance of various layers and generalization gap between the provided proxy samples and the real training data distribution, which leading the performance degradation. On the contrary, by applying the structural risk minimization principle, SlimLLaVA formulates the generalization gap of the pruning policy and refines pruning policy candidates to enhance task performance and generalization ability. Generally speaking, with an average pruning rate of 0.5, the presented search space evolution for pruning boost the accuracy by 2.6% (80.45% vs. 83.05%) and speed up $\times$ 2.16 compared by the none pruned model which get better result.

In order to demonstrate the generalization ability of our pruning policy, we also evaluate LLaVA-v1.5-7B across different dataset including ScienceQA, Vizwiz, MM-Vet and LLaVA-Bench. Compared with SparseGPT, with an average pruning rate of 0.5, SlimLLaVA still achieves 57.51% accuracy on ScienceQA, which proves the versatility of our method, achieving excellent performance. Moreover, in complex multimodal tasks, SlimLLaVA demonstrates superior generalization capabilities

Table 4: The accuracy on VizWiz, MM-Vet and LLaVA-Bench for LLaVA-v1.5-7B, where we evaluate the models across a wide range of domains.

| Method | Vizwiz | | | | MM-Vet | | | | LLaVA-Bench | | | |
|---|---|---|---|---|---|---|---|---|---|---|---|---|
| | 0 | 0.3 | 0.4 | 0.5 | 0 | 0.3 | 0.4 | 0.5 | 0 | 0.3 | 0.4 | 0.5 |
| LLM-Pruner | | 20.85 | 7.30 | 0.00 | | 14.30 | 0.10 | 0.00 | | 37.70 | 29.30 | 22.70 |
| SparseGPT | 50.05 | 46.43 | 41.90 | 38.61 | 35.40 | 21.20 | 20.30 | 17.90 | 55.90 | 57.90 | 52.50 | 48.80 |
| Ours | | 52.70 | 52.19 | 50.28 | | 33.80 | 33.00 | 31.70 | | 58.40 | 54.50 | 53.00 |

and adaptability, including Vizwiz(38.61% vs 50.28%), MM-Vet(17.90% vs 31.70%) and LLaVA-Bench(48.80% vs 53.00%) which pruning ratio is 0.5, underscoring its significant potential not only in content generation but also across a wide range of applications that require robust, context-aware processing. For pruned model with low pruning ratio outperforming dense model, we believe that removing ambiguous neurons during the pruning process can lead to better performance on some simple question-answering datasets.

Finally, we perform an investigation into how effectively sparse LVLM can be accelerated in practice using standard tools for CPU. Due to the limitation of experiment equipment, we can not use NVIDIA's official CUTLASS library which is supported by NVIDIA GPUs of generation Ampere and latest theoretically offering × 2 acceleration of matrix multiplications. Therefore, we investigate acceleration of unstructured sparsity for CPU inference based on DeepSparse. Figure 1 shows the detail of acceleration which can speed up ×1.47 for LLaVA-v1.5-7B while the accuracy degradation is only 0.45%. Although LLM-Pruner inference without depending on particular kernels, it is unacceptable of the decrease of performance. Compaerd with SparseGPT, SlimLLaVA achieve higher performance with the similar speed up. The observed speedups are nearly at the theoretical optimum, which suggests that unstructured sparsity acceleration for LLM inference on CPUs is already a practical approach.

**Visualization reasoning examples:** We present several qualitative visual reasoning examples of the LLaVA-SQA-7B model using the ScienceQA dataset in Figure 4. Compared to LLM-Pruner, SlimLLaVA obtains more accurate and specific answers to vision-language question pairs, which reveals the higher model capacity even with the similar sparsity. Although LLM-Pruner can produce accurate results in the first case, it also provide irrelevant wrong information. For SparseGPT, the generated table provides incorrect answers. SlimLLaVA can infer correct results and additionally analyzing relevant knowledge. In the second example, LLM-pruner hallucinates that melted marshmallow exists in the image leading to incorrect result. Conversely, SparseGPT accurately identifies the glass but made a critical error by overlooking the chocolate milkshake. SlimLLaVA demonstrates the ability to comprehensively analyze the attributes of all items presented in the task. By leveraging its multimodal reasoning capabilities, it can accurately interpret both visual and textual information to identify the relevant features of each object.

## 5 CONCLUSION AND FUTURE WORK

In this paper, we present SlimLLaVA, an automatically pruning method for LVLMs with limited sampled data, to achieve low latency and unaffected task performance. We institute the generalization gap of the pruning policy between the calibration samples and the unknown training set for the pruning policy using the structural risk minimization principle. By estimating via the Euclidean distance of the candidate policy set, we enhance the upper bound of the generalization ability for all policies in the pruning space. The pruning policy candidates evolve with the goal of optimizing task performance and generalization by refining the vision projector. SlimLLaVA focuses on minimizing the generalization gap, allowing it to generalize across various scenarios after a single search, without the need for retraining or task-specific fine-tuning. Extensive experiments on question-answer datasets demonstrate that SlimLLaVA achieves competitive performance compared to state-of-the-art pruned LVLMs.

While SlimLLaVA has significantly improved generalization ability and maintained the performance of pruned LVLMs, there are still areas for future work. We aim to develop more efficient search algorithms to reduce the search cost and design a pruning policy that reduces the activation matrix to achieve higher compression ratios, further improving inference time.

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
