# OpenReview forum: "SlimLLaVA: Automatic Pruning for Large Vision-language Models"
_ICLR.cc/2025/Conference — ICLR 2025 Conference Withdrawn Submission_

### Official Review · Reviewer_1RuU · 2024-10-30

**Soundness:** 3
**Presentation:** 2
**Contribution:** 3
**Rating:** 5
**Confidence:** 3

**Summary:**

The paper presents SlimLLaVA, a method for pruning large vision-language models for efficiency. SlimLLava proposes to only use a few samples to search the desired pruning policy by maximizing its generalization ability on the unknown training data. Vision projector is optimized by iteratively searching of the optimal pruning policy. Experiments on multimodal tasks like visual question answering show it maintains high accuracy while reducing model complexity, offering a 1.47x speedup over denser models on ScienceQA.

**Strengths:**

- SlimLLaVA provides an effective way to prune large vision-language models, using a few samples to assign the optimal pruning
ratio for weight matrix. The pruned model achieves a good trade-off between efficiency and accuracy.
- The proposed method can outperform existing methods on various datasets under different pruning ratio, which shows the effectiveness of SlimLLaVA.
- The ablation studies and analysis are thorough, which provides insights in different aspects of the proposed method.

**Weaknesses:**

- The authors mentioned many times in the paper that this pruning strategy is for efficient multimodal reasoning. However, from both the design space and evaluation datasets, it's not specific to reasoning. I.e., the design is not "optimized" for reasoning, and only ScienceQA in evaluation datasets are somewhat reasoning, and the other datasets are general recognition/perception benchmarks. This claim should be toned down.
- Only four datasets were evaluated. More extensive evaluation on different types of datasets are needed.
- The authors can also compare the pruning methods with some quantization methods. E.g., compare the accuracy under similar inference speed or vice versa.

**Questions:**

- Do authors find any drawbacks/failure cases of SlimLLaVA compared to previous method e.g., SparseGPT?
- Will the code be open-sourced?

---

### Official Review · Reviewer_MiUU · 2024-11-02

**Soundness:** 2
**Presentation:** 3
**Contribution:** 3
**Rating:** 5
**Confidence:** 4

**Summary:**

This paper presents an automatic pruning method for large vision-language models (LVLMs), aimed at reducing model complexity to facilitate deployment on resource-constrained devices. Specifically, it innovatively explores the objective of LVLM pruning from the perspective of model generalization capability and introduces the structural risk minimization principle to the searching process for optimal pruning policy. Meanwhile, the search space is iteratively updated and a weighted fitness function is designed to enhance model accuracy. The proposed method surpasses a range of existing LLM pruning approaches, demonstrating superior speed-accuracy trade-offs.

**Strengths:**

+The paper is clearly written, logically rigorous, and meets the general standards of academic writing.
+The issues of generalization and efficiency in pruning are indeed critical topics for the effective deployment of large models.
+The proposed method is straightforward and easy to implement. The simple introduction of a term similar to regularization to enhance the generalization capability of the pruning process is enlightening for me.
+Extensive experimental results demonstrate the effectiveness and generalization capability of the proposed method.

**Weaknesses:**

-The experiments are somewhat insufficient. a) Ablation studies are only performed on generalization considerations and the evolution of the search space, lacking a detailed analysis of specific designs, such as the application of Eq. (7). Or if it is not your original contribution, a citation for the source is needed. b) Lack of discussion on certain hyperparameter settings, including $\eta$ in Eq.(4) and $\tau$, $k$, and $n$ in Algorithm 1. A more detailed analysis will enhance the credibility of the method.
-The analysis of the experimental results appears somewhat disorganized. The speed up rate is mentioned prior to the discussion of the related experimental setup, yet no corresponding figures or tables are referenced. Additionally, the reported speed up effect is confusing. SparseGPT achieves a 2.16x speedup at 60% sparsity, while the proposed method reaches such result at only 50% sparsity. An explanation for this discrepancy would be appreciated.
-The writing could be improved, as some critical information is not sufficiently presented. a) There is a lack of theoretical analysis of the relationship between Rademacher complexity and matrix norms. In fact, simply citing previous work adds to the burden of reading and comprehension. b) Noting that the proposed method involves evolutionary algorithms to generate candidate pruning policies, the specific schemes employed are not adequately described.

**Questions:**

(1) Why is the evolutionary objective to maximize the fitness function, which comprises the loss function and matrix norm term, rather than minimizing it?
(2) For the implementation details, the meanings of "choices of pruning ratio" and "candidate pruning ratio" are unclear. What is the relationship between the them?
(3) As far as I know, the PTB dataset contains only texts. How is such data used for LVLM pruning? How is the loss function calculated?
(4) I want to discuss the consistency between the insight and the implementation. According to the stated motivation, the introduction of the matrix norm term in Eq.(4) aims to maximize the generalization ability of the proxy data over the training corpus. Given that the parameters of the LLM are frozen, it follows that as the pruning rate increases, this term gradually decreases. Does this imply that the proxy data's generalization ability increases along with the pruning rate? If so, why does SlimLLAVA exhibit a higher degradation rate than SparseGPT at high pruning rates, as shown in Tab.3?

---

### Official Review · Reviewer_EadW · 2024-11-03

**Soundness:** 2
**Presentation:** 1
**Contribution:** 2
**Rating:** 3
**Confidence:** 4

**Summary:**

The paper aims to search for the best pruning ratios for each layer by minimizing the model's loss and a norm regularization term, which the latter term is related to reducing the generalization errors. To further increase the capacity of the method, the paper's method updates the projector layers as well to minimize the model's loss. The method and baselines are evaluated on LLaVA and show the proposed method is better than SparseGPT and LLM-Pruner.

**Strengths:**

1. Connecting weights norm to the model's generalizability is interesting.

2. With the dynamic sparsity, the method outperforms SparseGPT.

**Weaknesses:**

1. The comparison to LLM-Pruner is not very fair. The proposed method is apparently for pruning the model in an unstructured manner (correct me if I am wrong), but LLM-Pruner is a structured pruning approach so the performance will not be comparable.

2. The method part is not straightforward to follow. Please see "Questions" for more details.

3. The background and related work for pruning on LVLM are lacking, and I think they should be discussed or cited. For example,

[1] ECoFLaP: Efficient Coarse-to-Fine Layer-Wise Pruning for Vision-Language Models

[2] Upop: Unified and progressive pruning for compressing vision-language transformers

[3] MoPE-CLIP: Structured Pruning for Efficient Vision-Language Models with Module-wise Pruning Error Metric

The ECoFLaP's idea is very related to this paper, so it would be great to include it in the experiments in the paper.

4. In lines 249-250, "However, calculating the upper bound for all pruning policies is computationally infeasible" is repeated with the previous sentence.

**Questions:**

1. In equation (6), how to know if the loss is upper bound or not?

2. In lines 254-255 and line 4 in Algorithm 1, what are the "candidates" mean? How to accurately find out what candidates are among the top k?

3. In Algorithm 1, does the pruning's performance or pruning policy keep improving through steps?

4. In line 357, does the "Auto w/o Gen" only remove the regularization term on weight's norm and keep the rest of the components the same?

5. How to optimize equation (6) to get the updated $W_p$? Is it done by gradient descent or evolution algorithm?

---

### Official Review · Reviewer_kyc8 · 2024-11-04

**Soundness:** 3
**Presentation:** 3
**Contribution:** 2
**Rating:** 6
**Confidence:** 4

**Summary:**

This paper proposed an evolution search based pruning method for vision language models.
The method works by identifying that the projector in typically vision language models like LLaVA plays an important role in pruning, and uses this projector to define a fitness function.
The experiments on LLaVA-SQA demonstrate the effectiveness of the proposed pruning methods.

**Strengths:**

1. The task of pruning for vision language models can have real-world applications.
2. The proposed method considers the architecture of vision language models, and makes it more suitable for pruning LVLMs.

**Weaknesses:**

1. The experiments are only conducted on a limited set of datasets and models. It seems that the only architecture used here is LLaVA?

**Questions:**

1. How does the proposed method generalizes beyond the LLaVA-v1.5 architecture? for example, what about models like MiniGPT4 or LLaVA-Next?
2. It seems that the similar techniques can also be used for models for audio-language models or even more modalities, including these could greatly enhance the paper.

---

### Note · Authors · 2024-11-13

I have read and agree with the venue's withdrawal policy on behalf of myself and my co-authors.